# Association of Maternal Longitudinal Hemoglobin with Small for Gestational Age during Pregnancy: A Prospective Cohort Study

**DOI:** 10.3390/nu14071403

**Published:** 2022-03-28

**Authors:** Shangzhi Xu, Weiming Wang, Qian Li, Li Huang, Xi Chen, Xu Zhang, Xiaoyi Wang, Weizhen Han, Xingwen Hu, Xuefeng Yang, Liping Hao, Guoping Xiong, Nianhong Yang

**Affiliations:** 1Hubei Key Laboratory of Food Nutrition and Safety, MOE Key Laboratory of Environment and Health, Department of Nutrition and Food Hygiene, School of Public Health, Tongji Medical College, Huazhong University of Science & Technology, Wuhan 430030, China; xushangzhi@shzu.edu.cn (S.X.); d202181615@hust.edu.cn (W.W.); union_qianli@hust.edu.cn (Q.L.); chenxi2917@163.com (X.C.); zhangxutjmu@icloud.com (X.Z.); xiaoyigood1990@163.com (X.W.); xxyxf@mails.tjmu.edu.cn (X.Y.); haolp@mails.tjmu.edu.cn (L.H.); 2Key Laboratory of Xinjiang Endemic and Ethnic Diseases (Ministry of Education), Department of Public Health, School of Medicine, Shihezi University, Shihezi 832003, China; 3Shenzhen Baoan Center for Chronic Disease Control, Shenzhen 518101, China; tjhuangli@126.com; 4Department of Obstetrics and Gynecology, The Central Hospital of Wuhan, Wuhan 430014, China; jewel_wh@163.com (W.H.); hyh0120@163.com (G.X.); 5Clinical Laboratory, Hubei Maternal and Child Health Hospital, Wuhan 430070, China; huxingwen@hbfy.com

**Keywords:** hemoglobin, hemoglobin change, SGA, birth outcomes, maternal

## Abstract

Background: Few studies have investigated the association of maternal longitudinal hemoglobin (Hb) with small for gestational age during pregnancy. The current study examined the associations of maternal Hb concentrations and Hb changes throughout the middle and late stages of pregnancy with small for gestational age (SGA) in a large prospective cohort study. Methods: This was a prospective cohort study, which enrolled pregnant women at 8–16 weeks of gestation and followed up regularly. Maternal Hb concentrations were measured at the middle (14–27 weeks) and late (28–42 weeks) stages of pregnancy, and the Hb change from the middle to late stage of pregnancy was assessed. The Log-Poisson regression model was used to identify the association of maternal Hb with SGA, including the implications of Hb during specific pregnancy periods and Hb change across the middle to late stages of pregnancy. Of the total 3233 singleton live births, 208 (6.4%) were SGA. After adjusting for potential confounders, compared with Hb 110–119 g/L, Hb ≥ 130 g/L at late pregnancy was significantly associated with a higher risk of SGA (adjusted RR: 2.16; 95% CI: 1.49, 3.13). When Hb changes from the middle to late stages of pregnancy were classified by tertiles, the greatest change in the Hb group (<−6.0 g/L) was significantly associated with a lower risk of SGA (adjusted RR: 0.56; 95% CI: 0.37, 0.85) compared with the intermediate group (−6.0~1.9 g/L). In conclusion, for women at low risk of iron deficiency, both higher Hb concentrations in late pregnancy and less Hb reduction during pregnancy were associated with an increased risk of SGA.

## 1. Introduction

Small for gestational age (SGA) is usually defined as a neonate birth weight lower than the 10th percentile for a specific completed gestational age by sex [1]. Accumulated evidence has suggested that SGA not only elevates risk of perinatal morbidity and mortality [2], but also relates to poor school performance in adolescents [3,4] and increased metabolic disease risk in adulthood [5]. It is therefore crucial to identify modifiable risk factors for SGA and implement effective preventive strategies.

Multiple determinants of SGA, including maternal underweight, malnutrition, poor placental transfer, and adverse environmental factors [6,7], have been identified. However, studies on the relationship between maternal hemoglobin (Hb) concentration and SGA have been inconsistent. Some studies have found that low maternal Hb concentration increases the risk of SGA, others have observed that both low and high maternal Hb concentrations were associated with increased odds of SGA [8,9,10]. These inconsistencies may stem from the difference in gestational age at which Hb was measured. Moreover, the relationship between Hb concentrations and SGA was usually examined separately during pregnancy in previous studies. Hb concentrations changing with gestation progresses due to the influence of physiological plasma volume expansion or hemodilution are an important physiological adaptation, which can facilitate utero–placental circulation [11]. Therefore, the association of maternal Hb concentration with SGA may be confounded by plasma expansion, and more studies focused on the change of Hb concentration during pregnancy are needed.

We aimed to explore the association of maternal Hb with SGA risk using data from a large prospective cohort study in Wuhan, China, considering maternal Hb concentration at both the middle and late stages of pregnancy, as well as the longitudinal change in maternal Hb from the middle to late stages of pregnancy.

## 2. Materials and Methods

### 2.1. Study Population

This study was embedded in the Tongji Maternal and Child Health Cohort (TMCHC), a prospective cohort study designed to examine the effects of maternal nutrition, lifestyle, and environmental indicators on maternal and child health in Wuhan, China [12]. From January 2013 to May 2016, women at 8–16 weeks of gestation who initiated prenatal care in the research hospitals were invited to join the cohort. Inclusion criteria were as follows: age ≥ 18 years, residing in Wuhan, and the ability to read and speak Chinese. All women were furnished with written informed consent at enrollment. The study was approved by the Ethics Review Committee of the Tongji Medical College of Huazhong University of Science and Technology in China (Code: Huazhong U-2013-02).

Participants in the present study were all from TMCHC, and the additional criteria were: (1) singleton live birth; (2) with Hb measurement in the middle and late stages of pregnancy. After excluding women who were lost to followup (*n* = 619), miscarriage (*n* = 222), multiple pregnancy (*n* = 167), and missing information on Hb (*n* = 52), a total of 3233 eligible mother–newborn pairs were included in the final analysis. Power calculation showed that a sample size of 1628 participants with at least 407 in each group would give us a power of 80% at the significance level of 0.05 to detect an RR of 2.

Previous studies suggested that gestational hypertension or preeclampsia was related to both higher Hb concentrations and SGA risk [13]. To avoid the potential influence, we excluded participants with gestational hypertension or preeclampsia for sensitivity analysis.

### 2.2. Measurement of Hb and Red Cell Indices

Maternal Hb and red cell indices measurements were performed during routine antenatal care. Data collected from medical records included Hb, mean corpuscular volume (MCV), mean corpuscular hemoglobin (MCH), mean corpuscular hemoglobin concentration (MCHC), and red cell volume distribution width (RDW). Anemia was defined as Hb < 110 g/L according to the WHO criteria [12]. Hb concentrations were categorized into four groups in each period (i.e., <110, 110–119, 120–129, and ≥130 g/L). There is substantial evidence demonstrating a U-shaped curve for the relation between maternal Hb and adverse birth outcomes [14], we chose Hb level at 110–119 g/L as the reference group. Hb change was calculated by using the Hb concentration at the late stage of pregnancy minus that at the middle stage of pregnancy. Hb change was categorized in tertiles, and the intermediate tertile was used as the reference group. Correspondingly, the time intervals between the two Hb measurements were calculated by using the gestational age of Hb measurement in the late stage of pregnancy minus that in the middle stage of pregnancy. 

### 2.3. Neonatal Outcomes

Information on birth outcomes was obtained from hospital medical records, including birth date, neonatal sex, birth weight, and length. SGA was defined as birth weight below the 10th percentile for a specific completed gestational age by sex using the Chinese neonatal birth weight criteria [15]. Gestational age was calculated by using the last menstrual period (LMP) self-reported by participants at the enrollment. If pregnant women could not accurately remember the LMP or reported menstrual disorders, gestational age was estimated by using fetal crown rump length measured through routine ultrasound examination in the early pregnancy. When the difference in the gestational age between the LMP and ultrasound was more than 10 days, the latter was chosen [16].

### 2.4. Covariates

Covariate information was collected via a face-to-face interview conducted by a trained investigator to complete a structured questionnaire at enrollment. Detailed information included maternal age at enrollment, ethnicity (Han Chinese, others), educational level (≤12, 13–15, ≥16 years), average personal income (per month; ≤CNY 4999, CNY 5000–9999, ≥CNY 10,000), gravidity (1, 2, ≥3), parity (primipara, multipara), abortion history (yes, no), LMP, alcohol consumption (yes, no), active or passive smoking (yes, no), iron supplement (yes, no). Pre-pregnancy weight was self-reported, and maternal height was measured by research personnel at enrollment. Maternal pre-pregnancy BMI was calculated as pre-pregnancy weight (kg) divided by height squared (m^2^). Gestational weight gain (kg) was calculated as the difference between the last weight measurement during pregnancy and the pre-pregnancy weight. Information on pregnancy complications included gestational hypertension (yes, no) and preeclampsia (yes, no) diagnosed by clinical doctor and obtained from medical records.

### 2.5. Statistical Analysis

Continuous variables were expressed by mean ± standard deviation (SD), and categorical variables were expressed by number and percentage (%). The two independent samples t-test was used to compare mean for continuous variables. The Pearson χ^2^ test was used to compare rate or proportion for categorical variables.

The Log-Poisson regression model was applied to estimate the risk ratio (RR) and 95% confidence interval (95%CI) for Hb concentration and its change with SGA, unadjusted and adjusted for maternal age at enrollment (continuous), height (continuous), ethnicity (Han Chinese, others), education level (categorical), average personal income (categorical), pre-pregnancy BMI (continuous), active or passive smoking (yes, no), alcohol consumption (yes, no), gravidity (categorical), parity (primipara, multipara), abortion history (yes, no), gestational weight gain (continuous), and neonatal sex (male, female). Model 3 further adjusted for iron-containing supplement (yes, no), gestational hypertension (yes, no), and preeclampsia (yes, no). In the analysis of Hb change, the time interval of Hb measurement (continuous) was additionally adjusted. 

All analyses were performed using the SPSS 22.0 for Windows (SPSS Inc., Chicago, IL, USA) with a 2-sided significance level of *p* = 0.05. 

## 3. Results

A total of 3233 eligible mother–infant pairs from the TMCHC study were included for final analysis. The baseline characteristics of the mothers and infants are shown in Table 1. In total, 208 (6.4%) of the newborns were identified as SGA. 

Hb concentration and red cell indices at the middle and late stages of pregnancy are shown in Table 2. The mean gestational ages at the time of the Hb measurements were 19.5 ± 4.0 week (middle stage of pregnancy) and 34.7 ± 3.8 week (late stage of pregnancy). The mean Hb levels were 116.7 ± 8.9 g/L at the middle stage of pregnancy and 116.2 ± 11.6 g/L at the late stage of pregnancy. Correspondingly, the rate of anemia increased as gestation progressed (from 19.3% in the middle stage of pregnancy to 27.7% in the late stage of pregnancy).

Table 4 presents the RRs and 95% CIs for the risk of SGA associated with Hb change from the middle to late stages of pregnancy, stratified by Hb in the late stage of pregnancy. In women with an Hb less than 130 g/L in late pregnancy, those with the greatest reduction from the middle to late stage of pregnancy (Hb change < −6.0 g/L) had a significantly decreased risk of delivering SGA infants (adjusted RR: 0.56; 95% CI, 0.37, 0.85) compared with women with an intermediate reduction (−6.0 ≤ Hb change ≤ 1.9). In women with Hb ≥ 130 g/L in late pregnancy, no significant association was found between longitudinal Hb change and SGA. There was no difference in the use of iron supplements among women within each stratified Hb group, but more women with Hb ≥ 130 g/L in late pregnancy had used iron supplements. 

## 4. Discussion

Previous studies demonstrated a U-shaped relationship between maternal Hb concentrations and adverse birth outcomes [14], but the relations differ by trimester. The link of low Hb concentration with SGA is more evident in early pregnancy and generally became weaker or nonexistent in the second or third trimester [9,17]. In line with previous evidences, no association between Hb concentrations in the middle stage of pregnancy and SGA was found in the present study. Evidence on associations between a high Hb concentration and adverse birth outcomes are mixed [18,19]. We observed that a higher Hb concentration in late pregnancy was associated with an increased risk of SGA. Little research has examined Hb changes from the middle to late stages of pregnancy with SGA. We highlighted this problem and observed that women with the greatest Hb reduction (equivalent hemodilution) from the middle to late stages of pregnancy (average Hb concentrations from 120.3 ± 7.8 g/L to 107.0 ± 8.9 g/L) were associated with a decreased risk of SGA. 

There are two potential mechanisms linking Hb concentrations and SGA risk. Firstly, higher Hb concentrations and lower Hb reduction may indicate poor plasma volume expansion, hemoconcentration [20], or high blood viscosity [21,22]. High blood viscosity can compromise placental blood flow, thereby reducing fetal oxygen and nutrition supply, restricting fetal growth and resulting in SGA [10,23,24,25]. In addition, some studies suggested that gestational hypertension or preeclampsia was independently responsible for both higher Hb concentrations and high SGA risk [26,27]. Our finding indicated that the association was independent of the two covariates by sensitive analyses excluding pregnant women with gestational hypertensive or preeclampsia. Secondly, association between greater Hb reduction from the middle to late stages of pregnancy and lower SGA risk may be explained by good maternal iron nutrition in early pregnancy and normal physiological plasma volume expansion or hemodilution. Hemodilution is an important physiological adaptation which can facilitate utero–placental circulation and may reduce SGA risk [11]. Previous studies have shown that low Hb in late pregnancy may more reflect normal hemodynamic adaptation and plasma volume changes rather than poor maternal nutrition [14,28,29]. 

Our study had several strengths. Firstly, the data were derived from a large prospective cohort study that allowed us to obtain Hb values from the middle to late stages of pregnancy and provided the possibility for Hb longitudinal research. In addition, Hb concentrations were evaluated prospectively, and the intervals of Hb measurements were identified. Secondly, our study adjusted important potential confounders, particularly gestational hypertension and preeclampsia, which have not been well controlled in other studies. We also adjusted the Hb measurement interval, so the bias due to measurements of Hb concentrations at different gestational weeks was minimized. Thirdly, from the perspective of epidemiological research, this work filled the gap between Hb change from the middle to late stages of pregnancy and SGA risk.

The study also had several limitations. Firstly, the majority of participants was Han Chinese and had moderate to high socioeconomic status with a low risk of iron deficiency, which might affect the generalizability of our results. However, the relative homogeneity of our study population was less likely to be affected by selection and confounding biases. Secondly, despite careful consideration of the known risk factors and potential confounders, residual confounding cannot be completely ruled out. Our data of pre-pregnancy body weight was self-reported; this is susceptible to recall bias, but these were thought acceptable as we collected the data at enrollment in early pregnancy when the body weight did not change significantly and the recall period was short. Thirdly, although we had a relatively large sample and chose the Hb value closest to the average gestational week in the middle and late stages of pregnancy to reduce bias, the fact that only one-time Hb measures occurred and a few women were lost to followup at delivery could lead the results to have a small bias. In addition, we only identified an association between Hb and SGA, further study is also warranted with regard to the potential mechanism for adverse effects of supplemental iron and high Hb concentrations during pregnancy. 

## 5. Conclusions

In conclusion, this study provides strong evidence that in women at low risk of iron deficiency, higher Hb concentrations in late pregnancy are significantly associated with increased risk of SGA. Greater Hb reduction from the middle to late pregnancy is significantly associated with decreased risk of SGA. Investigating the underlying clinical reasons of higher Hb concentrations and close followup with these women may help improve birth outcomes.

## Figures and Tables

**Table 1 nutrients-14-01403-t001:** Characteristics of the study population by SGA status ^†^.

Characteristics	Total (*n* = 3233)
Maternal characteristics	
Age (year)	28.3 ± 3.3
Height (cm)	160.4 ± 5.0
Pre-pregnancy weight (kg)	53.5 ± 7.4
Pre-pregnancy BMI (kg/m^2^)	20.8 ± 2.7
Educational level	
≤12 years	380 (11.8)
13–15 years	789 (24.4)
≥16 years	1966 (60.8)
Unclear	98 (3.0)
Average personal income	
≤4999 CNY	1149 (35.5)
5000–9999 CNY	1367 (42.3)
≥10,000 CNY	666 (20.6)
Unclear	51 (1.6)
Ethnicity (Han Chinese)	3151 (97.5)
Primipara	2735 (84.6)
Active or passive smoking (yes)	101 (3.1)
Alcohol consumption (yes)	54 (1.8)
Iron supplement (yes)	1747 (54.0)
Gestational weight gain (kg)	16.0 ± 4.5
GHD (yes)	176 (5.4)
Infant characteristics	
Gestational age (wk)	39.3 ± 1.4
Cesarean delivery	1315 (40.7)
Male	1753 (54.2)
Birth weight (g)	3343 ± 434
Length (cm)	50.2 ± 1.4
SGA *n* (%)	208 (6.4)

SGA, small for gestational age; BMI, body mass index; CNY, Chinese Yuan, 1 CNY ≈ 16 US$; GHD, include gestational hypertension and preeclampsia. ^†^ Data are presented as mean ± SD or *n* (%).

**Table 2 nutrients-14-01403-t002:** Characteristics of hemoglobin and red cell indices ^†^.

Characteristics	Mid-Pregnancy	Late Pregnancy	*p* Value
Gestational age at measurement (average, week)	19.5 ± 4.0	34.7 ± 3.8	
Hb (g/L)	116.7 ± 8.9	116.2 ± 11.6	0.034
Anemia	624 (19.3)	895 (27.7)	<0.001
Red blood cell count (×10^12^/L)	3.8 ± 0.4	3.9 ± 0.4	<0.001
Hct (L/L)	0.3 ± 0.1	0.3 ± 0.1	0.818
MCV (fl)	91.5 ± 5.0	92.8 ± 5.9	<0.001
MCH (pg)	30.4 ± 1.9	29.9 ± 2.3	<0.001
MCHC (g/L)	332.4 ± 11.4	321.9 ± 12.9	<0.001
RDW (%)	37.0 ± 12.9	35.7 ± 14.5	<0.001

Hb, hemoglobin; Hct, hematocrit; MCV, mean corpuscular volume; MCH, mean corpuscular hemoglobin; MCHC, mean corpuscular hemoglobin concentration; RDW, red cell volume distribution width. ^†^ Data are presented as mean ± SD or *n* (%) unless indicated otherwise. RRs and 95% CIs for the risk of SGA associated with Hb levels at the middle and late stages of pregnancy are shown in Table 3. After adjusting for potential confounders, compared with Hb 110–119 g/L, Hb ≥ 130 g/L at the late stage of pregnancy was significantly associated with a higher risk of SGA (adjusted RR: 2.16; 95% CI: 1.49, 3.13). However, neither middle stage pregnancy or late stage pregnancy anemia (Hb < 110 g/L) was found to be associated with SGA.

**Table 3 nutrients-14-01403-t003:** Associations between Hb concentrations in the middle and late stages of pregnancy and SGA.

Hb Concentrations (g/L)	*n*	SGA	Model 1 ^†^	Model 2 ^‡^	Model 3 ^§^
		[n (%)]	RR (95% CI)	aRR (95% CI)	aRR (95% CI)
Middle stage of pregnancy					
≥130	234	17 (7.3)	1.13 (0.68, 1.86)	1.30 (0.79, 2.15)	1.30 (0.79, 2.15)
120–129	997	64 (6.4)	0.99 (0.73, 1.36)	1.05 (0.77, 1.42)	1.05 (0.77, 1.42)
110–119	1378	89 (6.5)	Reference	Reference	Reference
<110	624	38 (6.1)	0.94 (0.65, 1.36)	0.86 (0.60, 1.23)	0.86 (0.60, 1.23)
Late stage of pregnancy					
≥130	396	45 (11.4)	1.96 (1.36, 2.81) *	2.19 (1.52, 3.15) *	2.16 (1.49, 3.13) *
120–129	842	59 (7.0)	1.21 (0.86, 1.70)	1.20 (0.85, 1.68)	1.19 (0.85, 1.68)
110–119	1100	64 (5.8)	Reference	Reference	Reference
<110	895	40 (4.5)	0.77 (0.52, 1.13)	0.70 (0.48, 1.03)	0.70 (0.48, 1.03)

Hb, hemoglobin; SGA, small for gestational age; RR, risk ratio; CI, confidence interval; aRR, adjusted risk ratio. ^†^ Model 1 unadjusted for any covariates. ^‡^ Model 2 adjusted for maternal age, ethnicity, education level, average personal income, pre-pregnancy BMI, active or passive smoking, alcohol consumption, parity, gestational weight gain, and newborn sex. ^§^ In the middle stage of pregnancy, Model 3 included Model 2 covariates and iron-containing supplement. In the late stage of pregnancy, Model 3 included Model 2 covariates, iron-containing supplement, gestational hypertension, and preeclampsia. * *p* < 0.001.

**Table 4 nutrients-14-01403-t004:** Associations between Hb changes from the middle to late stages of pregnancy and SGA stratified by Hb in late pregnancy.

	*n*	Hb-Middle	Hb-Late	Iron Supplement	SGA	Adjusted Model ^†^
		Mean ± SD	Mean ± SD	[*n* (%)]	[*n* (%)]	Adjusted RR (95% CI)
Hb-late < 130						
Hb change < −6.0 g/L	930	120.3 ± 7.8	107.0 ± 8.9	464 (49.9)	33 (3.6)	0.56 (0.37, 0.85) *
Hb change −6.0~1.9 g/L	901	116.7 ± 7.7	114.2 ± 7.8	479 (53.2)	60 (6.7)	Reference
Hb change ≥ 2.0 g/L	1006	110.9 ± 7.6	118.8 ± 7.3	552 (54.9)	70 (7.0)	1.08 (0.77, 1.50)
Hb-late ≥ 130						
Hb change < 8 g/L	121	130.4 ± 4.8	133.2 ± 2.6	73 (60.3)	11 (9.1)	0.75 (0.35, 1.59)
Hb change 8~15.9 g/L	140	123.2 ± 4.6	134.8 ± 4.2	93 (66.4)	17 (12.1)	Reference
Hb change ≥ 16 g/L	135	116.4 ± 6.6	139.0 ± 7.9	86 (63.7)	17 (12.6)	0.88 (0.47, 1.63)

Hb, hemoglobin; SGA, small for gestational age; RR, risk ratio; CI, confidence interval; aRR, adjusted risk ratio. ^†^ Adjusted Model adjusted for maternal age, height, ethnicity, education level, average personal income, pre-pregnancy BMI, active or passive smoking, alcohol consumption, parity, gestational weight gain, newborn gender, iron-containing supplement, gestational hypertension, preeclampsia, and Hb measurement interval. * *p* < 0.001. Sensitivity analyses were conducted to assess the robustness of the results. We excluded women with gestational hypertension or preeclampsia and found the results remained highly similar. (Appendix A).

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
