# Peer review of "Association of Maternal Longitudinal Hemoglobin with Small for Gestational Age during Pregnancy: A Prospective Cohort Study"

_nutrients, 2022, doi:10.3390/nu14071403_

Round 1

Reviewer 1 Report

The authors reported a longitudinal cohort study to examine the association between hemoglobin level during pregnancy and risk of small for gestational age in offspring. This manuscript was well written, and I only have some minor comments and suggestions for the authors. 

Methods

1. Page 2, Line 82

Please clarify the reason for choosing Hb level at 110-119 g/L as the reference group. 

2. Page 3, Line 105

The pre-pregnancy weight was self-reported. It will be a potential bias for study analysis. Please list this potential bias in the limitation part. 

3. Did the author collect the data on gestational diabetes in your cohort? Or also exclude women with GDM? 

Discussion

Page 7, Line 211. 

Due to the study design, the authors only could identify an association between Hb and SGA. But the causal inference from this observational study remains unclear. Please list this limitation in your manuscript.  

Reviewer 2 Report

Manuscript ID nutrients-1648356

Type Article

Title Association of maternal longitudinal hemoglobin during pregnancy with small for gestational age: A Prospective Cohort Study

This study aimed to explore the associations of maternal hemoglobin (Hb) with small for gestational age (SGA) risk using data from a prospective cohort study in Wuhan, China, considering maternal Hb concentration at both mid and late pregnancy, as well as the longitudinal change of maternal Hb from mid- to late pregnancy.

Comments and Suggestions for Authors:

The manuscript is an interesting study, but requires some considerations.

There is an error in the numbering of the pages of the manuscript, which makes it difficult to refer to them.

Line 16. In the Abstract section, the word hemoglobin must be followed by the acronym (Hb) and in the following citations put the acronym.

Line 71. It should be described whether a calculation of the sample size necessary to address the proposed objective was made a priori.

Line 73. The number of losses in the study was very high (n = 619). You should explain this issue.

Line 92. Gestational age was only estimated by using fetal crown rump length measured when pregnant women could not accurately remember the LMP. Didn't you do it in cases of menstrual frequency disorders?

Line 154. There is a typographical error. The header of Table 3 should be above Table 3.

Line 163. There is a typographical error. The header of Table 4 should be above Table 4.

Line 176. “In line with previous studies focused on the relationship between maternal Hb concentrations and SGA during pregnancy [9,16,17], we observed that higher Hb concentration in late pregnancy was associated with an increased risk of SGA. Meanwhile, no association between Hb concentrations in the mid-pregnancy and SGA risk was found”.  

However, in reference 9 we find: Ten studies with a sample size including 620,080 pregnant women entered the meta-analysis process. The overall relationship between maternal anemia during pregnancy and SGA was not significant (RR = 1.11 [95%CI: 0.99-1.24, p = .074]). The relationship between anemia during pregnancy and SGA based on pregnancy trimester showed that maternal anemia was significant in the first trimester, (RR = 1.11 [95%CI: 1-1.22, p = .044]), but this relationship was not significant in the second trimester (RR = 1.11 [95%CI: 0.85-1.18, p = .91]). Badfar et al. Maternal anemia during pregnancy and small for gestational age: a systematic review and meta-analysis. J Matern Fetal Neonatal Med. 2019 May;32(10):1728-1734. doi: 10.1080/14767058.2017.1411477.

And in reference 16 we can see: Raised Hb at 27-29 weeks gestation is associated with fetal growth restriction. Cordina et al. Association between maternal haemoglobin at 27-29 weeks gestation and intrauterine growth restriction. Pregnancy Hypertens. 2015 Oct;5(4):339-45. doi: 10.1016/j.preghy.2015.09.005.

Please clarify these aspects correctly.

Line 181. This paragraph should be removed from the Discussion section, as it only repeats results data.

Line 212. The authors honestly acknowledge the limitations of the study, especially in that it affects the generalizability of the results, but these limitations should be included in the Conclusions section. Thus, the results of this study with 3233 eligible pregnant women with very specific sociodemographic characteristics and therefore very little generalization (external validity), are contrary to the recent meta-analysis of Badfar et al. with 620,080 pregnant women.

The authors should reflect on whether presenting again data on the relationship between maternal Hb and SGD, this time from a very specific population, helps to resolve this old question, or would it be better to explore new ways that explain this relationship more clearly (hemorheological parameters ...).

Reviewer 3 Report

The study by Xu et al is fascinating, it is very well elaborated and includes a large population so the statistical evidence is robust, it is also very well narrated. It is surprising that an increase in Hb at the end of pregnancy is associated with SGA. The evidence demonstrates this association. 

  1. However, it is not clear to me the relevance of this article in a nutrition journal. Perhaps, it would be necessary to include a section supporting the nutritional need or supplements and the relationship to these data?
  2. In Table 2, it would be useful to include the p-value between groups and the p-value for dependent samples.
  3. Tables 3 and 4 need text in the results section. Also, it would be useful to include a symbol indicating the factor that was significant in the model, in addition to the 95%CI.

Minor comments:

  • Include at least 5 keywords.
  • the paragraph between lines 127 and 130 should be moved to section 2.1.
  • Review the author's guidelines for the format of the references.

Round 2

Round 2

The authors make changes in the new version that clarify the manuscript. However, there are still some aspects that should be addressed.

The fact that it is a population with low risk of iron deficiency should be included in the Abstract section.

Line 87. Appropriate references should be provided.

Line 155. In the text Table 2 should be without bold.

Line 160. In the table header, Table 3 should be in bold.

Line 166. In the text Table 4 should be without bold and remove the period after Table 4.

Line 173. In the table header, Table 4 should be in bold.

Line 247. The change in the letter source must be corrected.
